# The Authentication of Gayo Arabica Green Coffee Beans with Different Cherry Processing Methods Using Portable LED-Based Fluorescence Spectroscopy and Chemometrics Analysis

**DOI:** 10.3390/foods12234302

**Published:** 2023-11-28

**Authors:** Meinilwita Yulia, Analianasari Analianasari, Slamet Widodo, Kusumiyati Kusumiyati, Hirotaka Naito, Diding Suhandy

**Affiliations:** 1Department of Agricultural Technology, Lampung State Polytechnic, Jl. Soekarno Hatta No. 10, Rajabasa, Bandar Lampung 35141, Indonesia; meinilwitayulia@polinela.ac.id (M.Y.); analianasari@polinela.ac.id (A.A.); 2Spectroscopy Research Group (SRG), Laboratory of Bioprocess and Postharvest Engineering, Department of Agricultural Engineering, The University of Lampung, Bandar Lampung 35145, Indonesia; 3Department of Mechanical and Biosystem Engineering, Faculty of Agricultural Engineering and Technology, IPB University, Dramaga, Bogor 16680, Indonesia; slamet_ae39@apps.ipb.ac.id; 4Department of Agronomy, Faculty of Agriculture, Universitas Padjadjaran, Sumedang 45363, Indonesia; kusumiyati@unpad.ac.id; 5Department of Environmental Science and Technology, Graduate School of Bioresources, Mie University, 1577 Kurima-machiya-cho, Tsu-city 514-8507, Mie, Japan; naito@bio.mie-u.ac.jp; 6Department of Agricultural Engineering, Faculty of Agriculture, The University of Lampung, Jl. Soemantri Brojonegoro No. 1, Bandar Lampung 35145, Indonesia

**Keywords:** LED-based fluorescence spectroscopy, single-origin coffee, cherry processing method, PCA, LDA, PLS-DA, PCA-LDA

## Abstract

Aceh is an important region for the production of high-quality Gayo arabica coffee in Indonesia. In this area, several coffee cherry processing methods are well implemented including the honey process (HP), wine process (WP), and natural process (NP). The most significant difference between the three coffee cherry processing methods is the fermentation process: HP is a process of pulped coffee bean fermentation, WP is coffee cherry fermentation, and NP is no fermentation. It is well known that the WP green coffee beans are better in quality and are sold at higher prices compared with the HP and NP green coffee beans. In this present study, we evaluated the utilization of fluorescence information to discriminate Gayo arabica green coffee beans from different cherry processing methods using portable fluorescence spectroscopy and chemometrics analysis. A total of 300 samples were used (*n* = 100 for HP, WP, and NP, respectively). Each sample consisted of three selected non-defective green coffee beans. Fluorescence spectral data from 348.5 nm to 866.5 nm were obtained by exciting the intact green coffee beans using a portable spectrometer equipped with four 365 nm LED lamps. The result showed that the fermented green coffee beans (HP and WP) were closely mapped and mostly clustered on the left side of PC1, with negative scores. The non-fermented (NP) green coffee beans were clustered mostly on the right of PC1 with positive scores. The results of the classification using partial least squares–discriminant analysis (PLS-DA), linear discriminant analysis (LDA), and principal component analysis–linear discriminant analysis (PCA-LDA) are acceptable, with an accuracy of more than 80% reported. The highest accuracy of prediction of 96.67% was obtained by using the PCA-LDA model. Our recent results show the potential application of portable fluorescence spectroscopy using LED lamps to classify and authenticate the Gayo arabica green coffee beans according to their different cherry processing methods. This innovative method is more affordable and could be easy to implement (in terms of both affordability and practicability) in the coffee industry in Indonesia.

## 1. Introduction

Coffee is one of the most important crop products in the world, with Indonesia playing an important role as the fourth largest global coffee producer. Indonesia contributes an average annual production of 683.64 million kg per year (which is about 11%) to the world’s global coffee production [1,2]. In Indonesia, several expensive green coffee beans such as Gayo arabica coffee are produced in the Aceh province, which has a specific climate and unique cherry processing methods [2]. Gayo arabica coffee is also well known as one of the important specialty coffees with a high quality and which is traded at an expensive price [3]. There are three main regions in Aceh province for high-quality Gayo green coffee bean production sites, namely, Central Aceh, Bener Meriah, and Gayo Lues region, which contribute 28.23% to the total Indonesian coffee production [2]. In many areas in Indonesia, several different coffee cherry processing methods have been implemented, including dry or natural, wet (fully washed), and honey or semi-dry processes [4,5]. Fermented Gayo arabica green coffee beans, known as Gayo arabica wine coffee, are also recognized as important arabica green coffee beans from Aceh, with international recognition as the most expensive Gayo arabica coffee in the world. For this reason, the price of Gayo arabica green coffee beans is strongly influenced by the cherry processing methods used [4].

For accurate authentication of green coffee beans, several analytical methods based on physical, chemical, and optical properties have been well reported. Some physical features of green coffee beans such as color, shape, and size are directly related to high-quality beans. Using that information, an application of a computer vision system for quality and defect inspection of green coffee beans has been established, and 90% accuracy could be obtained [6]. This imaging system is easy to use, relatively low-cost, and a green technology where intact measurements of green coffee beans can be established. However, this method is not believed to be suitable for evaluating green coffee beans with similar physical properties such as the evaluation of several green coffee beans from the same variety [6]. High-performance liquid chromatography (HPLC) has been selected by many researchers as the most versatile targeted analytical method for assessing the quality of green coffee beans based on chemical properties [7,8]. The results are accurate. However, several drawbacks exist such as being expensive, time-consuming, laborious with the involved chemical-based sample preparation, and technically not available in most laboratories in developing countries such as Indonesia [9].

On the other hand, various spectroscopy-based analytical methods using optical properties as a potential marker are available. These could be used for the non-targeted discrimination of intact green coffee beans according to different types of beans, cherry processing methods, and beans’ geographical origin. Arabica green coffee beans with a better quality and selling at higher prices could be successfully discriminated from robusta green coffee beans using Raman spectroscopy, near-infrared (NIR) spectroscopy, and laser-induced breakdown spectroscopy (LIBS) [10,11,12,13,14]. The identification of coffee variety (mostly to separate between arabica and robusta green coffee beans) was also evaluated by using nuclear magnetic resonance (NMR) spectroscopy and mid-infrared (MIR) spectroscopy [15,16]. Several spectroscopy methods in different electromagnetic regions, from NIR to Terahertz (THz), have been used to authenticate green coffee beans according to different cherry processing methods [17,18,19,20]. In particular, recently, the UV-visible region (usually from 200 nm to 700 nm) has been popular as a simple spectroscopic tool for green coffee bean authentication, both through using absorbance and fluorescence spectral data [21,22,23,24,25]. Overall, spectroscopy-based methods for green coffee bean evaluation are acceptable, with fast spectral data measurement and relatively affordable instrumentation. Spectroscopy is mostly a green method that is environmentally friendly and leads to zero chemical waste as well as the possibility for intact and in situ spectral acquisition without any sample preparation.

Currently, the utilization of a portable spectrometer equipped with LED lamps as a light source for quality evaluation purposes has been raised due to its affordability and flexibility. However, there are also several drawbacks of LED-based spectroscopy, such as the spectral range limitation and the necessity of arranging multiple LEDs for multiple excitation wavelengths. LED-based fluorescence spectroscopy with multiple LED excitation lamps will have a low spectral resolution compared with conventional spectroscopy systems [26,27,28]. On the other hand, Suhandy et al. [29] reported an authentication method based on portable and single LED-based fluorescence spectroscopy to assess adulteration in stingless bee honey (SBH). Preliminary laboratory work on the application of LED-based fluorescence spectroscopy on instant coffee geographic discrimination has been reported [30]. However, according to the literature and to the best of our knowledge, the application of portable and single LED-based fluorescence spectroscopy for intact green coffee bean authentication has not yet been studied. For this reason, in this study, we aimed to evaluate a promising application of portable and single LED-based fluorescence spectroscopy to evaluate Gayo arabica green coffee beans according to their different cherry processing methods. Three classification methods based on PLS-DA, LDA, and PCA-LDA were used to perform supervised classification tasks.

## 2. Materials and Methods

### 2.1. Green Coffee Bean Samples

The Gayo green coffee bean samples were collected directly from Takengon, Central Aceh (4°35′39.0″ N 96°48′48.8″ E) (See Figure 1). The samples are Gayo arabica green coffee beans with three different cherry processing methods: honey process (HP), wine process (WP), and natural process (NP). The HP and WP are both fermented coffee and are recognized as a modification of the wet cherry processing method [31]. For HP, the selected red coffee cherries are first peeled to remove the skin and preserve the wet parchment and then dried. For WP, the selected red cherries are washed, put in a transparent polyethylene (PE) plastic container, and closed tightly for 12 days of fermentation. After 12 days, the coffee cherries are dried without direct exposure to sunlight [32]. The NP is the most implemented coffee cherry processing method [33]. Here, the selected red cherries are directly dried (unwashed), utilizing natural sun-drying or artificial drying. Dried cherries are then peeled to expose the non-fermented green coffee beans [33]. The different cherry processing methods resulted in green coffee beans with different physical properties, as seen in Figure 2. It is not an easy task to classify the HP, WP, and NP green coffee beans using their shape and color information. The moisture content of all samples (HP, WP, and NP) was kept at 12% according to the Indonesian National Standard for green coffee beans (SNI 01-2907-2008).

In this present study, one hundred (*n* = 100) samples of Gayo arabica green coffee bean samples were prepared for each cherry processing method (total *n* = 300 samples). For each sample, three non-defective green coffee beans were selected for spectral acquisition. The samples were stored at room temperature (28 °C) in a light-protected plastic container until the sample spectral data acquisition [34].

### 2.2. Spectral Acquisition

The intact spectral acquisition of Gayo arabica green coffee bean samples was conducted directly without any sample preparation using a low-cost, factory-calibrated, and portable fluorescence spectrometer from GoyaLab (Talence, France) equipped with 4 LED lamps (peak at wavelength 365 nm) as a light source, as seen in Figure 3. The use of a factory-calibrated portable spectrometer for food quality inspection purposes has been well reported [35,36]. The spectrometer was connected by a USB cable to the computer. The spectral acquisition was controlled by using SpectroLab software (version 1.0.24), and the parameter for fluorescence spectral data acquisition was determined as follows: exposure time 2000 ms and number of cycles 10. During spectral acquisition, the temperature and relative humidity of the experimental dark room were about 28 °C and 95%, respectively. For each sample, the fluorescence spectral data in the range of 348.5–866.5 nm with 0.5 nm interval were obtained for further chemometrics analysis.

### 2.3. Chemometrics Analysis

Unsupervised pattern recognition with 10 principal components (PCs) and a full cross-validation method were conducted based on principal component analysis (PCA). The result of PCA was generated for several plots such as score plot, x-loading plot, and influence plot. Three classification models based on PLS-DA, LDA, and PCA-LDA were used and compared. PLS-DA works based on a PLS regression algorithm. PLS-DA has been successfully applied, especially for evaluating green coffee beans, as currently reported in several published articles [12,18,21,22,37,38,39]. LDA and PCA-LDA have recently gained popularity as classic statistical methods for feature extraction and dimension reduction and are mostly employed among several supervised pattern recognition methods [40,41,42]. In LDA and PCA-LDA, the variance between the categories is to be maximized, and the variance within the categories is to be minimized [43]. To avoid model over-fitting in LDA and PCA-LDA models, it is required that the number of samples has to be at least twice as high as the number of variables, as mentioned by Harvey et al. [44]. In this study, all chemometrics were calculated by using the Unscrambler^®^ X version (10.5.1, CAMO, Oslo, Norway).

The comparison of classification model performance was evaluated based on the accuracy value. The accuracy rate of all classification models was determined by using the following equation [5,45]:(1)$$\mathrm{Accuracy}\;\mathrm{Rate}\;(\%)=\frac{\mathrm{Number}\;\mathrm{of}\;\mathrm{correct}\;\mathrm{classification}\;(N_{\mathrm{CORR}})}{\mathrm{Number}\;\mathrm{of}\;\mathrm{total}\;\mathrm{samples}\;(N_{\mathrm{TOT}})}\times 100\%$$

## 3. Results and Discussion

### 3.1. Typical Fluorescence Spectral Data

Figure 4 shows the typical broad fluorescence spectral data of Gayo arabica green coffee beans with three different cherry processing methods: the honey process (HP), wine process (WP), and natural process (NP). The obtained spectral data were highly noisy. After applying a moving average with 21-point smoothing, a better visualization of the fluorescence spectral data could be obtained, as seen in Figure 5. One major peak at around a wavelength of 460 nm was identified, and it was related to the presence of chlorogenic acids (CGAs), a polyphenolic compound in green coffee beans [46,47,48]. Suhandy and Yulia [48] measured the excitation–emission matrix (EEM) of several Indonesian ground-roasted coffees including peaberry and civet coffee. The highest fluorescence intensity was identified at 453 nm at a 370 nm excitation wavelength [48]. Several previous reports showed that green coffee beans are abundant with CGAs including caffeoylquinic acid (CQA), di-caffeoylquinic acid (diCQA), and feruloylquinic acid (FQA) [49,50,51]. The CGAs are important polyphenols in coffee and are widely used as indicators of the quality of coffee, since they directly affect the flavor and the nutritional value of the final coffee product [52]. The fermented green coffee bean samples (HP and WP) have a lower fluorescence intensity compared with those of the non-fermented green coffee beans (NP). This result is in line with several previous reports stating that the fermentation process is responsible for the decrease in the CGA content in green coffee beans [53,54]. A study on the influence of altitude, shade, and cherry processing method on CGAs in green coffee beans reported that unwashed coffee beans had significantly higher 3-caffeoylquinic (3-CQA) and caffeoyl feruloyl quinic acid (CFQA) contents than washed coffee beans [54].

A minor peak was observed at wavelengths around 670 and 720 nm both for fermented (HP and WP) and non-fermented (NP) green coffee bean samples. The peak at the wavelength of 670 nm is closely correlated to the presence of chlorophyll in green coffee beans, as is also reported in a previous work [30]. It is noted that the chlorophyll contents are not completely removed during the cherry processing method such as during the drying and fermentation process. Even after roasting, the fluorescence of the chlorophyll contents of ground-roasted coffee was also identified at a wavelength of 660 nm, as reported by Luo et al. [30]. A similar result was also observed in other green beans. For example, the chlorophyll fluorescence (CF) at wavelengths of 660 nm and 730 nm was also shown in soybean seeds, as demonstrated by Franca-Silva et al. [55].

### 3.2. PCA Analysis

PCA was calculated based on the smoothed fluorescence spectral data in the range of 348.5–866.5 nm. The result of PCA calculation using the first ten principal components (PC) is shown in Table 1, both for the calibration and validation steps. The scores of the first three principal components are depicted in Figure 6. The cumulative percentage variance (CPV) of PC1, PC2, and PC3 was 99% and met the required minimum CPV for reliable PCA [56]. The high CPV shows the quality of the analysis in transforming the original fluorescence spectral data into PCs. The result of the score plot showed that a possible separation of Gayo arabica green coffee beans according to different cherry processing methods could be established. The fermented (HP and WP) green coffee beans could be well separated from the non-fermented ones (NP). Along PC1 (with 91% explained variance), the non-fermented samples (NP) were mostly clustered in the right part with positive scores, while the fermented samples (HP and WP) were in the left part with negative scores. Along PC1, the proposed PCA is also enabled to discriminate two fermented green coffee beans between the HP and WP samples.

A plot with x-loadings in the y-axis and wavelengths in the x-axis is commonly used to analyze the most contributed variables during PCA calculation [57,58,59]. The high positive and negative x-loadings for the first three PCs are displayed in Figure 7, including the wavelengths at 390 nm, 470 nm, 480 nm, 680 nm, and 720 nm. This plot is important for investigating the most significant wavelengths that are responsible for the discrimination of fermented (HP and WP) and non-fermented (NP) green coffee bean samples. In PC1 and PC3, the fluorescence of CGAs could be identified at the wavelength of 470 nm and 480 nm in positive and negative directions. Since PC1 explained more than 90% of the spectral data variance compared with that of PC3, CGAs in Gayo arabica green coffee bean samples were mostly affected by PC1 only. In PC3, high x-loading was also identified at 680 nm, corresponding with the fluorescence of chlorophyll contents. Since the fermented coffee (NP) is mostly clustered at the right of PC1, it is suggested that the non-fermented coffee (NP) has higher CGAs compared with those of the fermented coffee samples (HP and WP). The chlorophyll content is similar for the fermented (HP and WP) and non-fermented (NP) coffee samples, since along PC3, all Gayo arabica green coffee bean samples clustered at positive scores. Thus, the CGAs are a good candidate marker to authenticate Gayo arabica green coffee beans with different cherry processing methods.

The calculation of PCA was also utilized to investigate any possible outlier present by analyzing the influence plot of Hotelling’s T^2^ statistics and F-residual values [60,61]. Any sample with high Hotelling’s T^2^ statistics and F-residual values outside the limit (the red dashed lines) would have been assigned as outliers and excluded from further analysis. The limit for Hotelling’s T^2^ statistics and F-residual is 3.88566 and 3.81405, respectively, as seen in Figure 8. Several NP samples have high Hotelling’s T^2^ statistics and exceed the limit. These samples are well described by the model. However, those samples may be influential to the model. There are no reported samples that exceeded the two limits in this study, as seen in Figure 8.

### 3.3. Classification Model Development Result Using PLS-DA, LDA, and PCA-LDA

To ensure robustness, all models were validated using an external validation sample set. For this purpose, for each class (HP, WP, and NP), the samples were randomly assigned into three sets, namely, the calibration sample set (*n* = 50), validation sample set (*n* = 30), and prediction sample set (*n* = 20). The first classification model was the PLS-DA model. This model was developed using all calibration sample sets (total *n* = 150) and validated using all validation sample sets (total *n* = 90). The number of latent variables (LVs) for the PLS-DA calibration model was determined by the Unscrambler. The suggested number of LVs is 4. The obtained accuracy for calibration and validation using the PLS-DA model is 83.33% (number of correct samples = 125 out of 150) and 88.89% (number of correct samples = 80 out of 90), respectively. For LDA, the wavelengths at 390 nm, 470 nm, 480 nm, 680 nm, and 720 nm (five variables) were used as predictors. The LDA model accuracy was 95.00%, 96.67%, and 93.33% for the linear, quadratic, and Mahalanobis methods, respectively. Improved accuracy was obtained by using PCA-LDA. The PCA-LDA model was developed using the first 10 PCs (CPV = 99.96% for both calibration and validation, respectively). The obtained accuracy was 96.25%, 98.75%, and 97.50% for the linear, quadratic, and Mahalanobis methods, respectively. The best classification model using PCA-LDA with a quadratic method is displayed in Figure 9.

The previously reported classification model for green coffee bean authentication (both in intact and powder samples) using other spectroscopy methods was comparable to our results. Raman spectral data in the range of m 200 cm^−1^ to 2000 cm^−1^ with a resolution of 6 cm^−1^ was obtained, and it was combined with several discriminant analyses (DA) to classify several coffee cultivars [11]. The lowest accuracy rate of 98.7% was obtained for the LDA model. Okubo and Kurata [12] reported the use of NIR spectroscopy to authenticate intact green coffee beans from seven different locations including Cuba, Ethiopia, Indonesia (Bali, Java, and Sumatra), Tanzania, and Yemen. The classification using the soft independent modeling of class analogy (SIMCA) method gave an accuracy rate of 73–100%. Similar NIR spectroscopy was applied to classify intact green coffee beans from two continents, Central–South America and Asia. The classification model was developed using interval PLS-DA (iPLS-DA). The best PLS-DA model was obtained with an accuracy rate of 98.6% both for calibration and cross-validation [17]. The LDA classification model was also used to discriminate between powder of green coffee beans of the *Coffea arabica* (arabica) and *Coffea canephora* (robusta) species from Java Island, Indonesia, based on ultraviolet–visible (UV-Vis) and NIR spectroscopy-based determination of caffeine and chlorogenic acid contents. Accuracies of 97% and 95% were obtained for UV-Vis spectroscopy and NIR spectroscopy, respectively [23]. A discriminant analysis (DA) with an accuracy rate of 95–100% was used to classify three arabica green coffee beans from Java Island, Indonesia (Java Preanger, Bondowoso, and Malang). Long wavelength NIR spectra were measured for intact green coffee bean samples in the range of 1000–2500 nm [62]. Bona et al. [63] reported the application of Fourier transform near-infrared (FT-NIR) spectroscopy combined with the support vector machine (SVM) classification method to differentiate several arabica green coffee beans from different regions in Brazil. The intact spectral acquisition was performed in the range of 1100–2498 nm at 2 nm intervals. The accuracy of SVM in calibration and validation was 99.59–100%. Currently, UV-Vis spectroscopy in the region of 230–450 nm was successfully applied to classify green coffee beans with different geographical origins in Brazil [22]. Three classification models were applied, including soft independent modeling of class analogy (SIMCA), data-driven SIMCA (DD-SIMCA), and one-class partial least squares (OCPLS). The accuracy was high, especially for the DD-SIMCA and OCPLS methods, where 93% accuracy could be reported.

### 3.4. The Result of Predictions

Table 2 shows the result of our prediction using the three developed classification models: PLS-DA, LDA, and PCA-LDA. The total prediction samples were 60 (*n* = 20 for HP, WP, and NP). In general, the classification result was acceptable for all classification models, with the lowest accuracy of 83.33% obtained for the PLS-DA model. Since PLS-DA is a full-spectrum-based technique, the analytical information overlaps, which interferes with the performance of the classification models in terms of accuracy [64]. The highest accuracy rate of 96.67% was obtained for the PCA-LDA model. The superiority of PCA-LDA was also reported in several previous qualitative studies [5,65]. Lasalvia et al. [66] briefly compared the performance of PCA-LDA and PLS-DA for the classification of simulated vibrational spectra. They showed that the PCA-LDA model seems a little better than the PLS-DA model for a slightly major accuracy and specificity in the classification of FTIR spectra. It is noted that the accuracy of LDA (95.00%) is quite similar to that of the PCA-LDA model (96.67%). The LDA model is developed using only five variables that are directly related to the CGAs and the chlorophyll contents in green coffee beans. This model is quite easy to understand compared with the PCA-LDA model.

## 4. Conclusions

Portable LED-based fluorescence spectroscopy combined with three classification methods based on PLS-DA, LDA, and PCA-LDA have been applied to assess the Gayo arabica green coffee beans with different cherry processing methods. Overall, our results show that it is possible to classify the samples into three different classes regardless of the type of classification model used. The wavelengths at 480 nm and 680 nm make important contributions to the classification of Gayo arabica green coffee beans with different cherry processing methods. Those wavelengths are directly related to the CGAs and chlorophyll content in green coffee beans. This fluorescence method is easy to use in many Indonesian coffee industries with the following features: non-destructive, affordable, and fast without any sample preparation (possible for intact spectral acquisition). Additionally, to validate our result, it is also important to include other sources of sample variability by using coffee samples from other geographical origins, botanical origins (robusta and liberica), and different farming systems (organic and non-organic).

## Figures and Tables

**Figure 1 foods-12-04302-f001:**
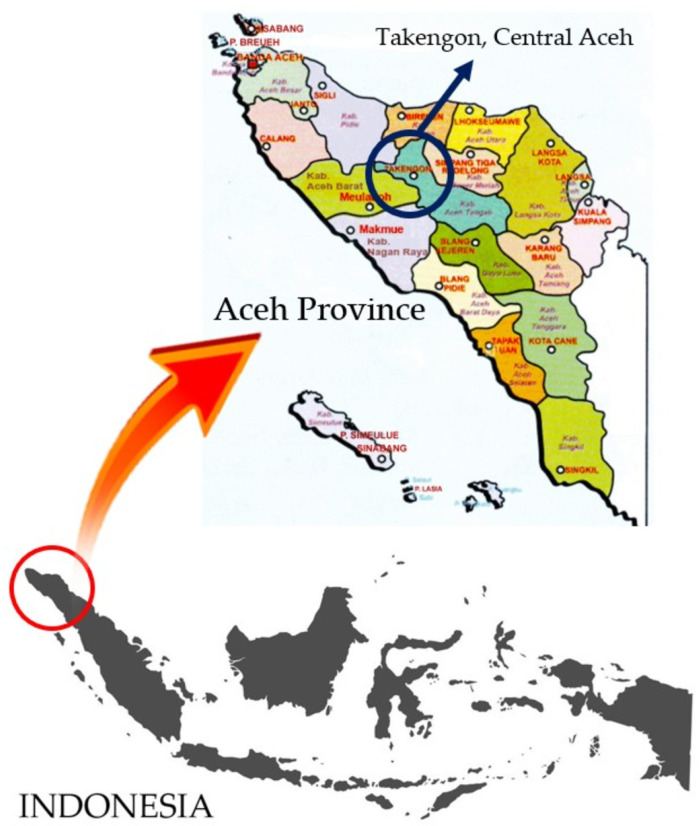
The geographical origin of Gayo arabica green coffee beans used in this study.

**Figure 2 foods-12-04302-f002:**
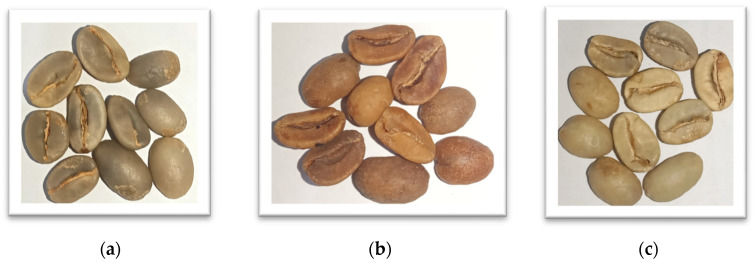
Three types of Gayo arabica green coffee beans were used in this study: (**a**) honey process (HP); (**b**) wine process (WP); and (**c**) natural process (NP).

**Figure 3 foods-12-04302-f003:**
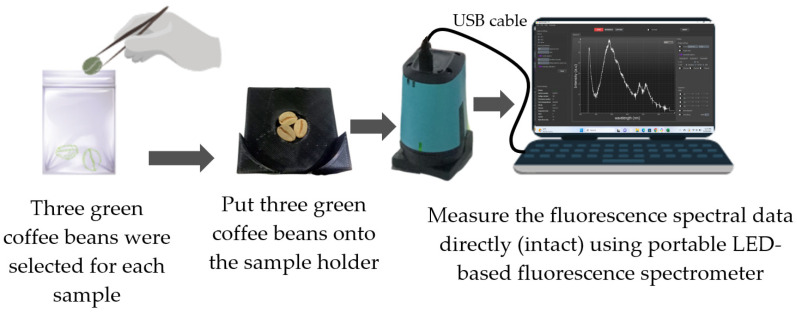
The intact spectral acquisition system for Gayo arabica green coffee beans using portable LED-based fluorescence spectroscopy.

**Figure 4 foods-12-04302-f004:**
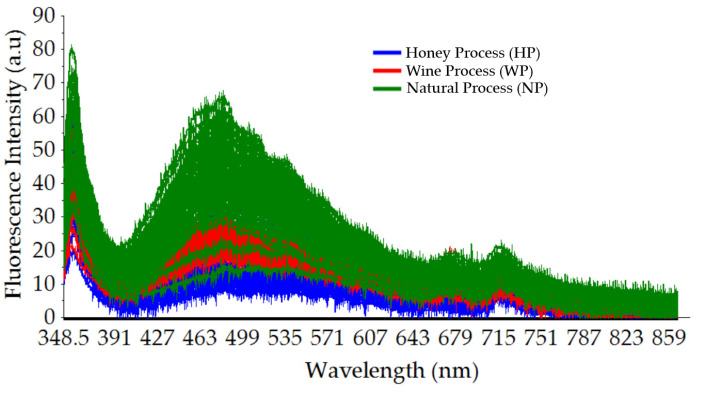
Raw typical fluorescence spectral data of intact Gayo arabica green coffee beans (HP, WP, and NP) with different cherry processing methods in the range of 348.5–866.5 nm.

**Figure 5 foods-12-04302-f005:**
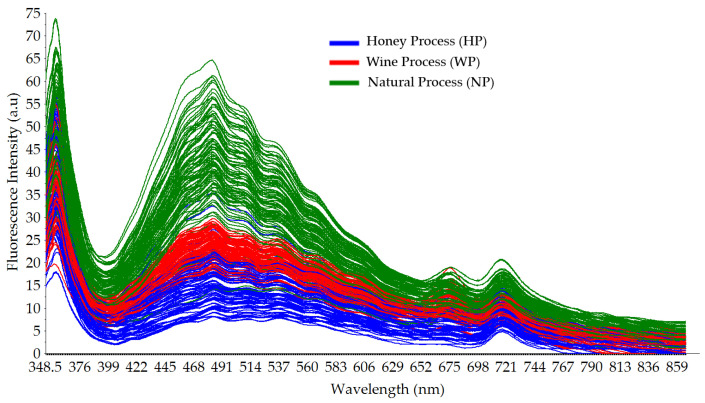
Smoothed typical fluorescence spectral data of intact Gayo arabica green coffee beans (HP, WP, and NP) with different cherry processing methods in the range of 348.5–866.5 nm.

**Figure 6 foods-12-04302-f006:**
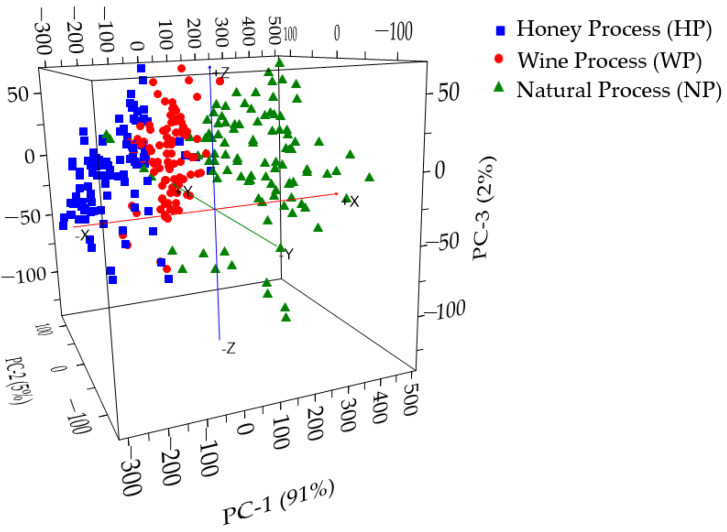
The score plot of the first three PCs from PCA calculation on the smoothed fluorescence spectral data in the range of 348.5–866.5 nm.

**Figure 7 foods-12-04302-f007:**
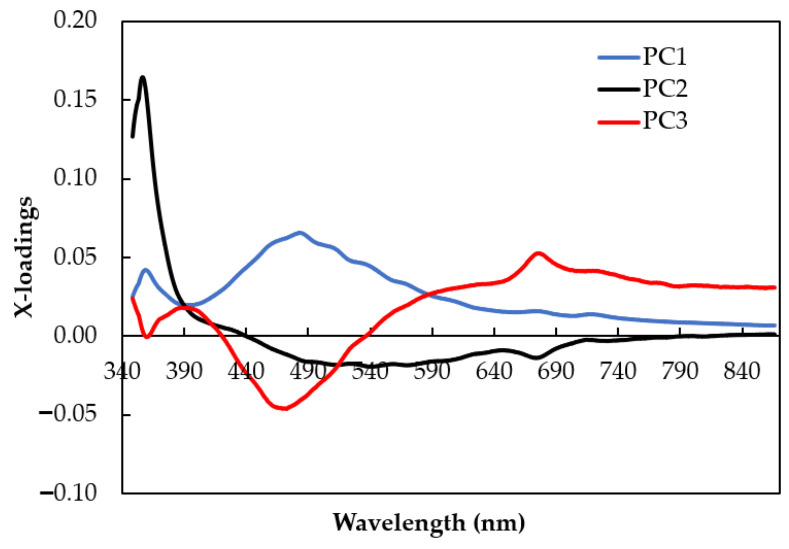
X-loading versus wavelength plot of the first three PCs from PCA calculation on the smoothed fluorescence spectral data in the range of 348.5–866.5 nm.

**Figure 8 foods-12-04302-f008:**
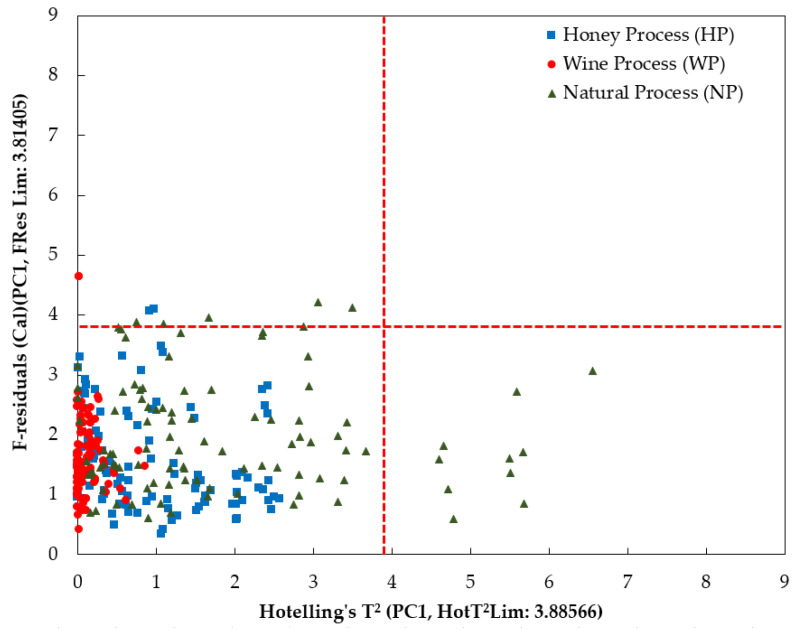
The influence plot from PCA calculation on the smoothed fluorescence spectral data in the range of 348.5–866.5 nm.

**Figure 9 foods-12-04302-f009:**
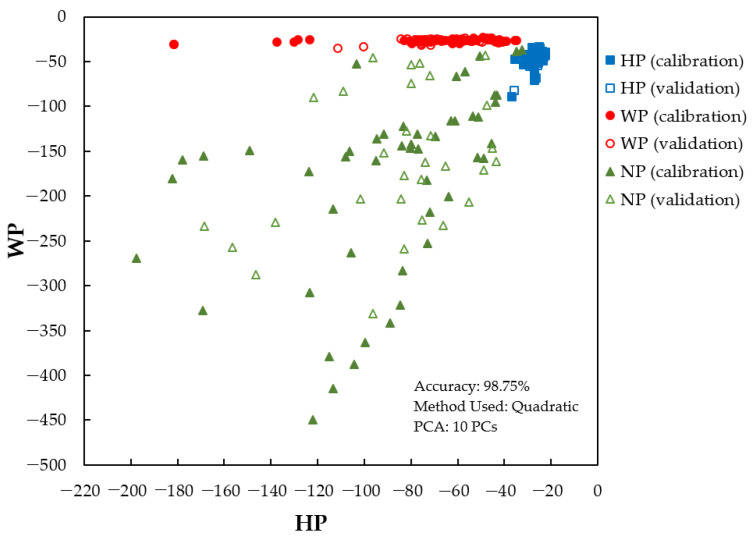
The PCA-LDA model with quadratic method calculated on the smoothed fluorescence spectral data in the range of 348.5–866.5 nm.

**Table 1 foods-12-04302-t001:** The result of PCA calculation using the first ten principal components (PCs) in the calibration and validation steps.

Principal Components (PCs)	Cumulative Percentages Variance (CPV) (%)	
	Calibration	Validation
PC1	91.16	91.00
PC2	96.14	96.03
PC3	98.58	98.52
PC4	99.30	99.27
PC5	99.56	99.53
PC6	99.73	99.71
PC7	99.88	99.87
PC8	99.92	99.92
PC9	99.95	99.94
PC10	99.96	99.96

**Table 2 foods-12-04302-t002:** The result of classification using the smoothed fluorescence spectral data in the range of 348.5–866.5 nm for three different classification models.

Classification Model		Samples	Actual			Accuracy Rate (%)
			HP	WP	NP	
PLS-DA	Predicted	HP	16	0	0	83.33
		WP	4	20	6	
		NP	0	0	14	
LDA	Predicted	HP	20	2	0	95.00
		WP	0	17	0	
		NP	0	1	20	
PCA-LDA	Predicted	HP	20	0	0	96.67
		WP	0	18	0	
		NP	0	2	20	

## Data Availability

The datasets generated for this study are available upon reasonable request from the corresponding authors.

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
