# Peer review of "The Authentication of Gayo Arabica Green Coffee Beans with Different Cherry Processing Methods Using Portable LED-Based Fluorescence Spectroscopy and Chemometrics Analysis"

_foods, 2023, doi:10.3390/foods12234302_

Round 1

Reviewer 1 Report

Comments and Suggestions for Authors

The Authentication of Gayo Arabica Green Coffee Beans with Different Bean Processing Methods Using Portable LED-Based Fluorescence Spectroscopy and Chemometrics Analysis

Abstract

1.      As in the 2nd line of abstract you mentioned different processing methods kindly briefly mention the significance of differentiating among green coffee beans processing methods for the quality of Gayo Arabica coffee (in just one sentence)

2.      At the end there are no understandable results for readers. Provide a short summary of the main findings, highlighting the clustering of fermented and unfermented green coffee beans on PC1. Summarize the key findings and emphasize the practical applications, particularly the affordability and practicability for the Indonesian coffee industry

3.      By addressing these points, you can provide a more reader-friendly abstract that effectively communicates the significance and findings of your study

Introduction

4.      In the line 53 write High quality Gayo green beans instead of Gayo high quality to avoid grammar mistake

5.      Mention the direct influence of processing methods on the price of Gayo arabica green coffee beans

6.      The sentence in line 78 is a bit long. Consider breaking it into two sentences for better readability.

7.      In lines 89-91, where you mention the acceptability of spectroscopy-based methods, provide a bit more detail on what "green method" refers to

Material and method

8.      According to my understanding and previous data heading (2.1. Green Coffee Bean Samples) was about sampling of green coffee beans. Why you mentioned red cherry in different lines such as in line 114 and 117. Consider using consistent terminology throughout. For example, in line 109, you refer to “Gayo arabica green coffee beans”, while in lines 112 and 115, you mention “red coffee cherries”. Clarify the terminology for better consistency.

Conclusion

9.      Kindly mention any potential future directions or applications of your research. This could include further refinement of the method, application to other coffee varieties or exploration of additional features for classification

Author Response

Dear reviewer,

We highly acknowledge your valuable comments and suggestions on our manuscript. Herewith we kindly sent you our replies.

Reviewer 2 Report

Comments and Suggestions for Authors

1. Although the paper provides a good description of the research methodology, it is recommended that more details on aspects such as spectrometer calibration and coffee bean sample preparation be provided in order to further improve the reproducibility of the study.

2. It is recommended that a comparative analysis be conducted with other existing coffee bean identification methods. This would help to highlight the advantages and potential limitations of portable LED fluorescence spectrometry.

3. A more extensive statistical validation of the results is recommended. Including additional statistical tests or cross-validation methods would enhance the robustness of the results.

Comments on the Quality of English Language

Overall, the English language quality of this paper is high. It meets the expected standards for a scientific publication and effectively communicates the research findings to the intended audience. More thorough checking for typographical or linguistic errors is recommended.

Author Response

Dear reviewer,

We are glad to receive your reviews. We have revised the manuscript according to your valuable comments and suggestions. Please kindly find our replies in the attachment file. Thank you

Best regards,
